# Microvascular Thrombosis as a Critical Factor in Severe COVID-19

**DOI:** 10.3390/ijms24032492

**Published:** 2023-01-27

**Authors:** Patricia P. Wadowski, Benjamin Panzer, Alicja Józkowicz, Christoph W. Kopp, Thomas Gremmel, Simon Panzer, Renate Koppensteiner

**Affiliations:** 1Division of Angiology, Department of Internal Medicine II, Medical University of Vienna, 1090 Vienna, Austria; 2Department of Medical Biotechnology, Faculty of Biophysics, Biochemistry and Biotechnology, Jagiellonian University, 30-387 Krakow, Poland; 3Department of Cardiology, Wilhelminenspital, 1160 Vienna, Austria; 4Institute of Antithrombotic Therapy in Cardiovascular Disease, Karl Landsteiner Society, 3100 St. Pölten, Austria; 5Department of Internal Medicine I, Cardiology and Intensive Care Medicine, Landesklinikum Mistelbach-Gänserndorf, 2130 Mistelbach, Austria; 6Department of Blood Group Serology and Transfusion Medicine, Medical University of Vienna, 1090 Vienna, Austria

**Keywords:** capillaries, coronavirus disease 2019, glycocalyx, microcirculation, platelets, thromboinflammation, thrombosis

## Abstract

Platelet–endothelial interactions have a critical role in microcirculatory function, which maintains tissue homeostasis. The subtle equilibrium between platelets and the vessel wall is disturbed by the coronavirus disease 2019 (COVID-19), which affects all three components of Virchow’s triad (endothelial injury, stasis and a hypercoagulable state). Endotheliitis, vasculitis, glycocalyx degradation, alterations in blood flow and viscosity, neutrophil extracellular trap formation and microparticle shedding are only few pathomechanisms contributing to endothelial damage and microthrombosis resulting in capillary plugging and tissue ischemia. In the following opinion paper, we discuss major pathological processes leading to microvascular endothelial activation and thrombosis formation as a possible major adverse factor driving the deterioration of patient disease course in severe COVID-19.

Coronavirus disease 2019 (COVID-19) emerged recently as a burden of global scope deteriorating healthcare systems [1,2,3]. COVID-19 is a prothrombotic and proinflammatory disease, with many immune mechanisms resembling other diseases, in particular (viral) infections, but with distinct changes in pathophysiological reactions [4,5]. Herein, it promotes systemic hyper-inflammation with an increased risk of thromboembolism, and the occurrence of microthrombosis is discussed as a major driver of disease severity [6]. The thrombotic response in COVID-19 patients involves all components of Virchow’s triad, including endothelial injury, stasis and a hypercoagulable state [7,8,9]. As a central mechanism of viral infection, glycocalyx shedding is discussed as being a critical and accelerating factor for viral entry (Figure 1) [10,11]. The latter can cause direct tissue injury such as endotheliitis and myocarditis [12,13,14]. Indirect cellular lesions are mediated by the immune response and hypercoagulability, where thromboinflammation is a key driver [14,15]. Microthrombi are regarded to be an underlying cause of myocyte necrosis in COVID-19 [16]. These thrombi are different in comparison to the composition of thromboemboli in COVID-19 negative subjects, having higher deposition of fibrin and the complement factor C5b-9 [16,17]. Deposits of the complement system such as C5b-9 and C4d were detected in the microvasculature of the lungs and skin [18]. Moreover, skin samples showing characteristic changes of severe COVID-19 could help to identify individuals at risk [17]. Microthrombus formation could therefore also explain the severe disease courses of COVID-19 in patients with pre-existing cardiovascular illness [19], which are especially prone to microcirculatory complications, as systemic microvascular perfusion is much lower than in healthy volunteers [20,21,22].

Endothelial cell damage due to the viral infection results in subendothelial collagen exposure initiating von Willebrand factor (vWF) binding via its A3 domain [24,25]. This promotes platelet adhesion via the glycoprotein (GP) Ib-V-IX complex, leading to platelet activation and aggregation and the induction of the coagulation cascade (further details are discussed later in the text) [26,27,28]. Platelet activation can be propagated by several agonists, the strongest of which is thrombin [29]. Thrombin binds to the protease-activated receptors (PAR) 1 and 4, and also to GP Ib, while collagen binds to the GP VI receptor on platelets [29,30]. This causes platelets to become activated, resulting in a prothrombotic cascade. In the first step of processes, arachidonic acid is converted into thromboxane A_2_, a vasoconstrictive agent, which has further potent proaggregatory properties [31,32]. These aforementioned influences cause the platelets to degranulate. The released dense granules are rich in ADP, which binds to the P2Y_12_ and P2Y_1_ receptors increasing platelet activation [27,33]. Alpha granules fuse with the cell membrane and release further procoagulatory and proinflammatory factors [33,34,35]. Among these factors is P-selectin, which binds to neutrophils, monocytes and other proinflammatory cells [36]. In neutrophils, the interaction with P-selectin also promotes the formation of neutrophil extracellular traps (NETs) [37]. P-selectin is expressed on activated platelets or shed into the circulation by the latter and endothelial cells [15,38]. The shedding of platelet adhesion receptors is a modulator of thromboinflammatory processes, also during COVID-19 [39].

Platelets undergo shape changes as a result of calcium mobilization and the dephosphorylation of vasodilator-stimulated phosphoprotein, causing physical platelet aggregation [40]. GP IIb/IIIa receptors link to GP IIb/IIIa receptors on other platelets making the bonds between platelets stronger [41]. This also causes nearby platelets to become activated by a mechanism called outside-in signaling [41]. Among others, vascular endothelial cells, platelets and blood cells are capable of secreting extracellular vesicles [42,43]. These vesicles contain RNA (e.g., microRNA and mRNA), cytokines, growth factors, lipids and small amounts of DNA [43,44]. Some of these extracellular vesicles expose phosphatidylserine, which has an up to 100 times greater procoagulatory property than activated platelets [45].

The other main component in thrombosis is the activation of the coagulation cascade either by tissue factor (TF; extrinsic pathway) or the intrinsic pathway [27]. Both pathways are active in COVID-19 [46,47,48] and result in factor X activation, causing thrombin generation [27]. Thrombin cleaves soluble fibrinogen and turns it into insoluble fibrin [27]. Fibrin forms interlocked strands, and this complex is stabilized by factor XIII [27]. As previously mentioned, thrombin activates platelets, but platelets play a role in activating the production of thrombin by membrane scramblase action [49,50].

Thrombin activates platelets even in subnanomolar concentrations [51]. Herein, PAR-1 is first activated, and as the thrombin concentration rises, PAR-4 activation is needed, being the predominant pathway for acceleration of platelet activation [51,52]. The signaling pathways of the PARs are still activated in patients with platelet inhibition by dual antiplatelet therapy consisting of aspirin and a P2Y_12_ receptor antagonist despite adequate P2Y_12_ receptor inhibition [53,54,55]. Other platelet activation pathways, which also play a role during SARS-CoV-2 infection, are toll-like receptors (TLRs), which are not targeted by current antiplatelet therapy [56,57].

TLR2 has been shown to play a role in sensing the E-protein of ß-coronaviruses, and blocking of TLR2 signaling resulted in a survival-benefit of SARS-CoV-2 infected mice [56]. The TLR2 pathway promotes platelet hyperreactivity and accelerates thrombosis [58]. Interestingly, TLR2 signaling upregulates the release of vWF from Weibel–Palade bodies of endothelial cells [59], and from alpha granules in megakaryocytes/platelets [60,61] vWF has a central role in microthrombosis formation during COVID-19 [62]. After sensing of the SARS-CoV-2 spike protein, TLR2 dimerizes with TLR1 or TLR6 and activates the proinflammatory NF-κB pathway [63].

TLR4 has a major role in the inflammatory response by getting activated by lipopolysaccharides (LPS). During SARS-COV-2 infection; however, the spike protein directly binds to the extracellular domains of TLR1, TLR4 and TLR6, having the strongest affinity to TLR4 [64]. Therefore, a strong inflammatory effect and platelet activation through pattern recognition of the SARS-CoV-2 spike protein by TLR4 is discussed [65] as being potentially one of the main pathophysiological pathways of microthrombosis in COVID-19. Furthermore, SARS-CoV-2 is suggested to infect enterocytes and hereby induce a disruption of the gut barrier [66]. This promotes bacterial translocation, which can be measured by increased LPS levels [66]. Consequently, platelet TLR4 activation by LPS during inflammation promotes platelet–neutrophil interactions, which induce endothelial damage and lead to NET formation [67]. Hereby platelets remain the main limiting factor in NETosis and the process occurs primarily in liver and pulmonary capillaries [67]. However, further studies on COVID-19 are necessary to explore detailed pathomechanisms.

An important fact in understanding potential therapeutic approaches is that pathways, which are not targeted by dual antiplatelet therapy (DAPT), among those PAR-1, collagen and epinephrine signaling still account for stable platelet aggregate formation [68]. The above-discussed observations might be the underlying pathomechanisms, which explain that there was no clear benefit of antiplatelet therapy in the prevention of thrombotic events in COVID-19 patients [69,70,71]. Therapeutic anticoagulation using heparin seems to be most effective in non-critically ill COVID-19 patients [72], but not in those with critical illness [73], and further strategies to overcome the processes of immunothrombosis are needed. However, patient outcome with regard to different anticoagulation treatment regimens warrants further clinical trials [72,73,74]. In addition, impaired fibrinolysis has emerged as another challenging therapeutic aspect, requiring future research [75,76,77,78]. Interference with thrombin-induced pathways could also be of potential benefit; however, the occurrence of increased bleeding events might limit therapeutic success [79,80]. Moreover, anticoagulation regimens after recovery from COVID-19 to overcome the higher risk of deep vein thrombosis and pulmonary embolism are yet not well studied and should also consider a higher bleeding risk in the first months after SARS-CoV-2 infection [81].

The inflammatory properties associated with COVID-19 infection add to the activation of platelets and the induction of the coagulation cascade. Leukocytes are recruited to the sites of inflammation, causing the release of cytokines (such as interleukin (IL) 6 and 8) among other proinflammatory substances [82,83]. Acute inflammation is associated with increased levels of fibrin strand density [84]. Increased thrombin generation promotes leukocyte recruitment and augments inflammation [85].

The upregulation of tissue factor in different cells during COVID-19, including endothelial and epithelial cells, neutrophils, monocytes and macrophages as well as platelets, links inflammation and thrombosis [46,48,86,87]. Tissue-factor-bearing extracellular vesicles are increasingly released into the circulation and activate the extrinsic coagulation pathway [48]. Extracellular vesicle tissue factor activity has been shown to be associated with thrombotic events and disease severity, circulating leukocytes and inflammatory markers, prothrombin time and d-dimer levels, vWF, ADAMTS13 (a disintegrin-like metallopeptidase with thrombospondin motif type 1 member 13), fibrinogen and plasmin–antiplasmin complexes [48,88,89]. In addition, contact activation of the coagulation system is triggered by neutrophil and complement activation [47,48].

During systemic inflammation endothelial function is deteriorated as a result of elevated vWF, which is released from Weibel–Palade bodies upon endothelial activation and from platelet alpha granules [33]. vWF enhances platelet-to-endothelium and platelet-to-platelet binding [33]. The activity of vWF is influenced by the molecular size and ultra-large vWF (UL-vWF) highly promotes thrombosis [90]. The maintenance of the equilibrium of vWF and its specific cleaving protease ADAMTS13 is a limiting factor in the process of microthrombosis, and the equilibrium is disturbed in systemic inflammation such as also severe COVID-19 [62,90,91]. A higher vWF activity is linked to in-hospital mortality of infected patients [26,91]. Moreover, a decrease in ADAMTS13 activity was associated with lower survival [91]. Therefore, as therapeutic measures caplacizumab (a bivalent single-domain antibody targeting the A1 domain of the vWF), recombinant ADAMTS13 and N-acetyl cysteine, which is known to reduce the size of vWF multimers, were suggested [26]. Autopsy results of patients with severe COVID-19 have shown the occurrence of severe capillary congestion, in part by microthrombi, some of which exhibit vWF positivity [62,92]. After release, vWF assembles into long strands and forms scaffolds to attach NETs [15,93]. Long vWF multimers and herein especially UL-vWF exhibit prothrombotic effects, contributing to glycocalyx disintegration by disturbing glycocalyx integrity, as vWF associates with heparan sulfate of syndecan-1 [62,94]. Multimeric vWF binds also to NO synthase, thereby decreasing NO availability during hypoxic conditions [95]. Moreover, NETs interact electrostatically with vWF, and this causes the retention of NETs on the endothelial surface [96,97]. In turn, a massive tissue injury is induced, which could be reduced by nearly 80% after blocking of vWF in mice [97]. One of the primary components of NETs, histones, directly induce endothelial damage during inflammation [97,98]. Histones exhibit a high content of positively charged amino acids and therefore can easily interact with the polyanions of the glycocalyx [99]. Histone accumulation is a major process driving microvessel dysfunction in acute lung injury. Hence, the design of polyanions to neutralize circulating histones was suggested to prevent acute lung injury [99]. Furthermore, histones and the DNA of NETs activate the coagulation system through promotion of thrombin generation [93]. The latter can occur through multiple pathways such as, among others, the induction of platelet activation by TLR2 and TLR4 through NETs [93]. Furthermore, histones are also capable of inducing platelet microaggregate formation in a GPIIbIIIa- independent, but fibrinogen-dependent manner [93]. The clinical benefits of heparin administration in, e.g., non-critically ill COVID-19 patients [72] might also be explained by the prevention of histone-platelet interactions, which is achieved by high heparin concentrations and was shown to protect mice from histone-induced tissue damage, thrombocytopenia and death [100].

One of the linking elements between vWF and NET formation is possibly IL-6, having a central role in the cytokine release syndrome of COVID-19 [91,101]. IL-6 is suggested to orchestrate a feedback mechanism between NETs and the vWF/ADAMTS13 axis [91]. The application of the IL-6 receptor antagonists tocilizumab and sarilumab showed to improve survival of critically ill COVID-19 patients [102]. In addition, NETosis is triggered by other cytokines, e.g., IL-1β and IL-8 [103,104]. Blockage of IL-1β by canakinumab was recently tested in patients with SARS-CoV-2 infection [105]. The ‘Canakinumab in Patients With COVID-19 and Type 2 Diabetes’ (CanCovDia) trial included 116 diabetic patients with COVID-19, which were randomized into two groups with canakinumab vs. placebo in addition to standard treatment [105]. Although canakinumab did not result in an improvement in survival, length of ICU stay or hospitalization as well as ventilation time and glycemic control was more rapidly achieved, requiring less treatment with antidiabetic drugs [105]. As shown in this trial, blocking of IL-1β might be of most advantage for patients at high risk for severe COVID-19 disease course, including those with diabetes-related complications, kidney failure and adiposity [105].

The inflammation induced by excessive NET formation might explain the progress of cardiovascular diseases during COVID-19 and other acute infections [15]. Herein, NETs promote intimal smooth muscle cell apoptosis and drive the destabilization of atherosclerotic plaques [106]. The latter promotes further changes in local hemodynamics and hypercoagulability [107].

Unfolding of vWF leads to exposure of the A2 domain, which exhibits high sequence homology with the complement factor B [62]. This enables the binding of activated complement C3b initiating the alternative complement pathway [62]. In turn, the formation of tissue-factor-enriched NETs is promoted [62].

Interference with the activity of ADAMTS13 is also conferred by thrombospondin 1 and platelet factor 4, which are secreted from activated platelets [62]. Both molecules bind to the vWF A2 domain and cause steric hindrance for ADAMTS13 binding, which prevents the cleavage of vWF [108,109].

Another pathophysiological aspect leading to higher NET release in COVID-19 is the occurrence of autoantibodies [110]. Herein, eight different types of autoantibodies targeting phospholipids and phospholipid-binding proteins (aPL antibodies) were detected during SARS-CoV-2 infection [110]. These aPL antibodies occurred in about half of the measured patients, and although assumed to be transient, they exhibited a prothrombotic potential by triggering neutrophil activation and NET release [110]. The autoimmune response leading to aPL antibody formation is complexly orchestrated by genetic predisposition, previous antigen exposures and the hyperactivation of the host immune system in response to environmental triggers, in this case SARS-CoV-2 infection [110,111].

The persistence of neutrophil activation and NET formation seems to be involved in the development of long COVID [112]. These changes can be observed even months after SARS-CoV-2 infection [112], and viral persistence is suggested to trigger repeated inflammatory responses [113]. In post-COVID-associated changes, neutrophils are discussed to promote fibrosis by protease release and to drive pulmonary symptoms [112]. In addition, NETosis is discussed to contribute to tumor growth, neurological disorders and worsening of concomitant diseases (e.g., hypertension or diabetes) after SARS-CoV-2 infection [104]. Given the involvement of NETs in (micro)thrombosis and vasculitis [114,115], it can be speculated that NETosis promotes post-COVID endotheliopathy and procoagulability. Recently, there is increasing evidence for microthrombus formation as a contributing factor to cardiopulmonary long COVID syndromes [116]. Moreover, as during acute SARS-CoV-2 infection, key factors of post-COVID sequelae are based on endothelial injury, inflammation and procoagulability [116,117]. Virus-eliminating strategies might be helpful in the prevention of persistent COVID-associated complications [118].

Prevention of NET formation may be achieved by application of dipyridamole, an antiplatelet agent, which interferes with aPL antibody-mediated NET release by adenosine A_2A_ receptor agonism [119]. Likewise, ticagrelor might exert its beneficial inhibitory effects on NET formation independently of P2Y_12_ inhibition by increasing adenosine levels [120]. NET release can also be abrogated by the JAK 1/2 inhibitor ruxolitinib, which is applied in myeloproliferative neoplasms [121]. Inhibitors of the complement system are further therapeutic options interfering with NET production and with the first reports of symptom improvement during COVID-19 [6,122,123].

NET degradation can be induced by dornase alpha, a recombinant DNase I, which can be inhaled and the application may reduce the occurrence of acute respiratory distress syndromes (ARDS) [6]. However, similarly to DNase, the administration of heparin was shown to dismantle NETs and to remove platelet aggregates from NETs [124].

However, future research should also emphasize preservation models of endothelial integrity, which provides a shield against thromboinflammatory reactions [10]. The endothelial surface layer and in particular the glycocalyx is a very fragile structure, which is very prone to adverse environmental changes [125]. Conformational changes of the glycocalyx can be conferred by proinflammatory stimuli, such as increased plasma Na^+^ levels [126,127]. These changes may also initiate pathological processes facilitating infections and immunothrombosis. Many clinical concepts for glycocalyx preservation are hardly feasible; however, some of those such as administration of albumin, antithrombin III, sulodexide, atrasentan (a selective endothelin A receptor antagonist), steroids or statins should be studied further [128,129,130,131,132,133,134,135,136].

In conclusion, endothelial dysfunction with glycocalyx disintegration can be regarded as an initial trigger of SARS-CoV-2-mediated microthrombosis. Signaling pathways contributing to immunothrombosis are complex, involving innate and adaptive mechanisms. COVID-19-associated prothrombotic potential relies on well-published findings, such as endothelial cell infection [13] with glycocalyx destruction, promoting subsequent collagen exposure with platelet and coagulation activation [137,138]. Microthromboses upon platelet–platelet and platelet–leukocyte interactions further augment this response [31,32,139]. These are specific therapeutic targets to interfere with prolonged procoagulant and prothrombotic responses to viral infection. Evidence for long COVID treatment is still scarce, although many aspects of the endothelial-based pathologies of the acute SARS-CoV-2 infection may apply to the long-term sequelae [116,117,118]. Research on the elimination of viral persistence as well as interference with processes of immunothrombosis is still ongoing and merits further study.

## Figures and Tables

**Figure 1 ijms-24-02492-f001:**
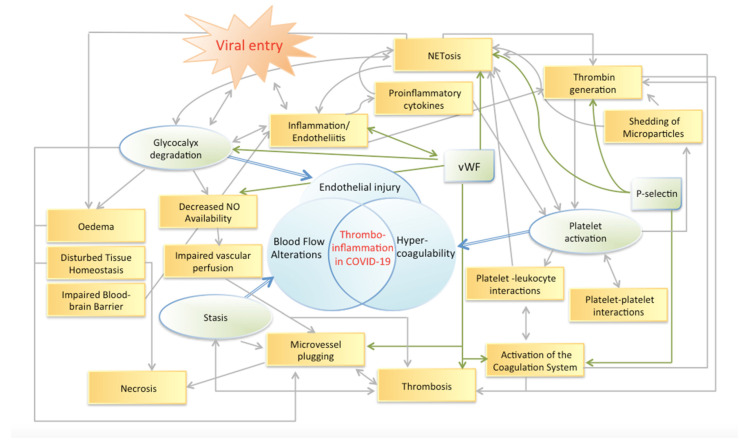
Endothelial dysfunction with glycocalyx degradation is postulated as the central mechanism induced through and facilitating viral entry. The resulting inflammatory and procoagulative processes initiate a complex cascade destabilizing the equilibrium of Virchow’s triad resulting in (micro-) thrombosis and tissue necrosis. The concept of the figure is modified after Ataga et al.—‘Hypercoagulability and thrombotic complications in hemolytic anemias.’ [23].

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
