# Peer review of "Microvascular Thrombosis as a Critical Factor in Severe COVID-19"

_ijms, 2023, doi:10.3390/ijms24032492_

Round 1

Reviewer 1 Report

In the present opinion article, Wadowski and her six coauthors present yet another article on the connection between COVID-19 and thrombosis. Personally, I would have preferred an article on the potential impact of microvascular thrombosis in the context of Long COVID, as I see this as the more relevant health threat (with way less literature). However, the authors obviously aim to present their concept on microvascular thrombosis as critical factor in COVID-19 severity. The article reads well, but several key reviews are not taken into account and the plasmatic coagulation is virtually absent (see below). Hence, I recommend revising this manuscript accordingly.

Major comments:

1)  Arguably it is hard to keep track with all of the COVID-19 literature (189 hits for “COVID AND thrombosis” alone for the last year), however, I was a bit astonished that of the close to 100 cited publications only third was from the last two years. At least some landmark reviews like Conway et al. Nat Rev Immunol 2022, Gorog et al. Nat Rev Cardiol 2022, Flaumenhaft et al. Blood 2022, Ahamed & Laurence J Clin Invest 2022 or Jonigk et al. Virchows Arch 2022 should be taken into account.

2)  Personally, I would have started with vWF-GPIb and not with GPVI-collagen. However, there are also arguments for the authors’ order of events. Nevertheless, I would recommend to at least mention vWF exposure on damaged endothelium (and on exposed collagen) in line 50 (with referring to more details later in the article).

3)  Line 90-95: I would recommend pointing out that the release of vWF following TLR2 signaling is mostly happening in endothelial cells. As the authors speak of TLR2 signaling in platelets just before this, one might assume that vWF released from platelets would be critical (however, at least to my knowledge there is no evidence for this).

4)  In line 104 it could be added that a role for TLR4 in SARS-CoV-2 responses remains to be determined, as there is no LPS in a classical COVID-19 infection that would drive NET formation (by the way I would prefer the term NET formation over NET production – the authors could write NETosis the second time in line 106 to avoid having NET formation twice).

5) Given the general title of this manuscript I am missing a paragraph on the effect of a SARS-CoV-2 infection on the plasmatic coagulation system and how this contributes to the elevated occurrence of microvascular thrombosis.

6) In my understanding of the terms “thrombo-inflammation” (the interplay of platelets and the immune system in mediating cellular responses, e.g. in the context of ischemic stroke) and “immunothrombosis” (a thrombotic event resulting from an overshooting immune response) I would say that the authors should replace thromboinflammation with immunothrombosis in line 206 and 212.

Minor comments:

1)  Of course, 2019/2020 is still quite recent, however, given the disruptive effects the COVID-19 pandemic has had on all of us, I am not sure whether “novel” coronavirus disease (abstract and introduction) is still a valid term. I would simply delete the “novel”.

2)  Besides degranulation (line 58ff.), one could mention receptor shedding as an additional mechanism, which affects thrombo-inflammation.

3)  Line 70: I would rephrase “vascular cells”, as at least for me it is not entirely clear what is meant here.

4)  vWF is first mentioned in line 93, but the abbreviation comes in line 122 – this should be changed.

Author Response

To the Editor-in-Chief

Vienna, 19 January 2023

Dear Professor Dear Professor Dr. Maurizio Battino,

Dear Professor Dr. Isabella Russo,

Dear Professor Dr. Maria Otilia Drugan,

We would like to submit a revised version of our manuscript, entitled Microvascular thrombosis as a critical factor in severe COVID-19“ (Manuscript ID: ijms-2136466), for publication in the International Journal of Molecular Sciences.

We would like to thank you and the referees for the review and the valuable comments and suggestions. The manuscript has been amended accordingly. We believe that these changes have resulted in a greatly improved manuscript, which we hope is now suitable for publication in International Journal of Molecular Sciences.

Yours sincerely,

Priv.-Doz. Patricia Wadowski, MD, PhD

Point-by-point response to Reviewer 1:  Thank you very much for carefully reading our manuscript and your valuable suggestions, which we have followed.

In the present opinion article, Wadowski and her six coauthors present yet another article on the connection between COVID-19 and thrombosis. Personally, I would have preferred an article on the potential impact of microvascular thrombosis in the context of Long COVID, as I see this as the more relevant health threat (with way less literature). However, the authors obviously aim to present their concept on microvascular thrombosis as critical factor in COVID-19 severity. The article reads well, but several key reviews are not taken into account and the plasmatic coagulation is virtually absent (see below). Hence, I recommend revising this manuscript accordingly.

Major comments:

1)  Arguably it is hard to keep track with all of the COVID-19 literature (189 hits for “COVID AND thrombosis” alone for the last year), however, I was a bit astonished that of the close to 100 cited publications only third was from the last two years. At least some landmark reviews like Conway et al. Nat Rev Immunol 2022, Gorog et al. Nat Rev Cardiol 2022, Flaumenhaft et al. Blood 2022, Ahamed & Laurence J Clin Invest 2022 or Jonigk et al. Virchows Arch 2022 should be taken into account.

Authors’ response:

Thank you for your valuable suggestion. We have now included the recommended references:

- Conway et al. Nat Rev Immunol 2022- reference number 8 of the revised manuscript

- Gorog et al. Nat Rev Cardiol 2022- reference number 47 of the revised manuscript

- Flaumenhaft et al. Blood 2022- reference number 9 of the revised manuscript

- Ahamed & Laurence J Clin Invest 2022 - reference number 115 of the revised manuscript

- Jonigk et al. Virchows Arch 2022- reference number 5 of the revised manuscript

In addition, we have added among others some further recent publications:

- Nguyen et al. Sci Rep 2022 (reference number 85 of the revised manuscript)

- Sachetto, Mackman Curr Drug Targets 2022 (reference number 86 of the revised manuscript),

- Zhu, Y., Chen, X., & Liu, X. (2022). Front Immunol, 13. (reference number 103 of the revised manuscript)

- Wang C et al. Frontiers in Cellular and Infection Microbiology. 2022;12. (reference number 116 of the revised manuscript)

- Root-Bernstein, R., Huber, J., & Ziehl, A. Int J Mol Sci, 2022; 23(19). (reference number 136 of the revised manuscript)

2)  Personally, I would have started with vWF-GPIb and not with GPVI-collagen. However, there are also arguments for the authors’ order of events. Nevertheless, I would recommend to at least mention vWF exposure on damaged endothelium (and on exposed collagen) in line 50 (with referring to more details later in the article).

Authors’ response: We have now revised the sentence accordingly.

Endothelial cell damage due to the viral infection results in subendothelial collagen exposure initiating von Willebrand factor (vWF) binding via its A3 domain.[1,2] This promotes platelet adhesion via the glycoprotein (GP) Ib-V-IX complex, leading to platelet activation and aggregation and the induction of the coagulation cascade (further details are discussed later in the text).[3-5]

This is now mentioned on page 2, lines 67-71 of the revised manuscript.

3)  Line 90-95: I would recommend pointing out that the release of vWF following TLR2 signaling is mostly happening in endothelial cells. As the authors speak of TLR2 signaling in platelets just before this, one might assume that vWF released from platelets would be critical (however, at least to my knowledge there is no evidence for this).

Authors’ response: Thank you for this comment. We have amended the mentioned sentence.

TLR2 signaling upregulates the release of vWF from Weibel-Palade bodies of endothelial cells,[6] and from alpha granules in megakaryocytes/ platelets.[7,8]

This is now mentioned on page 3, lines 121-123 of the revised manuscript.

4)  In line 104 it could be added that a role for TLR4 in SARS-CoV-2 responses remains to be determined, as there is no LPS in a classical COVID-19 infection that would drive NET formation (by the way I would prefer the term NET formation over NET production – the authors could write NETosis the second time in line 106 to avoid having NET formation twice).

Authors’ response: We have now revised the paragraph on TLR4 - SARS-CoV-2 interactions:

The TLR4 has a major role in the inflammatory response by getting activated by lipopolysaccharides (LPS). During SARS-COV-2 infection, however, the spike protein directly binds to the extracellular domains of TLR1, TLR4 and TLR6, having the strongest affinity to TLR4. [9] Therefore, a strong inflammatory effect and platelet activation through pattern recognition of the SARS-CoV-2 spike protein by TLR4 is discussed [10] being potentially one of the main pathophysiological pathways of microthrombosis in COVID-19. Furthermore, SARS-CoV-2 is suggested to infect enterocytes and hereby induce a disruption of the gut barrier.[11] This promotes bacterial translocation, which can be measured by increased LPS levels.[11] Consequently, platelet TLR4 activation by LPS during inflammation promotes platelet-neutrophil interactions, which induce endothelial damage and lead to NET formation.[12] Hereby platelets remain the main limiting factor in NETosis and the process occurs primarily in liver and pulmonary capillaries. [12] However, further studies in COVID-19 are necessary to explore detailed pathomechanisms.

This is now mentioned on page 3, lines 127-140 of the revised manuscript.

5) Given the general title of this manuscript I am missing a paragraph on the effect of a SARS-CoV-2 infection on the plasmatic coagulation system and how this contributes to the elevated occurrence of microvascular thrombosis.

Authors’ response: We have now included a paragraph on the plasmatic coagulation system effects of COVID-19.

The up-regulation of tissue factor in different cells during COVID-19 including endothelial and epithelial cells, neutrophils, monocytes and macrophages as well as platelets links inflammation and thrombosis.[13-16] Tissue factor bearing extracellular vesicles are increasingly released into the circulation and activate the extrinsic coagulation pathway.[15] Extracellular vesicle tissue factor activity has been shown to be associated with thrombotic events and disease severity, circulating leukocytes and inflammatory markers, prothrombin time and d-dimer levels, vWF, ADAMTS13 (a disintegrin-like metallopeptidase with thrombospondin motif type 1 member 13), fibrinogen and plasmin–antiplasmin complexes.[15,17,18] In addition, contact activation of the coagulation system is triggered by neutrophil and complement activation. [19,15]

This is now mentioned on page 3, lines 164-171 and page 4, lines 184-186 of the revised manuscript.

6) In my understanding of the terms “thrombo-inflammation” (the interplay of platelets and the immune system in mediating cellular responses, e.g. in the context of ischemic stroke) and “immunothrombosis” (a thrombotic event resulting from an overshooting immune response) I would say that the authors should replace thromboinflammation with immunothrombosis in line 206 and 212.

Authors’ response: We have revised the term according to your recommendation (line 310 and 316 of the revised manuscript).

Minor comments:

1)  Of course, 2019/2020 is still quite recent, however, given the disruptive effects the COVID-19 pandemic has had on all of us, I am not sure whether “novel” coronavirus disease (abstract and introduction) is still a valid term. I would simply delete the “novel”.

Authors’ response: Thank you for the comment - we have now deleted the word “novel” before coronavirus disease (line 21 and 32 of the revised manuscript).

2)  Besides degranulation (line 58ff.), one could mention receptor shedding as an additional mechanism, which affects thrombo-inflammation.

Authors’ response: We now mention the mechanism of receptor shedding as a contributing factor to thromboinflammation.

The released dense granules are rich in ADP, which binds to the P2Y12 and P2Y1 receptors increasing platelet activation.[20,4] Alpha granules fuse with the cell membrane and release further pro-coagulatory and pro-inflammatory factors.[21,22,20] Among these factors is P-selectin, which binds to neutrophils, monocytes and other pro-inflammatory cells.[23] In neutrophils, the interaction with P-selectin also promotes the formation of neutrophil extracellular traps (NETs).[24] P-selectin is expressed on activated platelets or shed into the circulation by the latter and endothelial cells.[25,26] The shedding of platelet adhesion receptors is a modulator of thromboinflammatory processes, also during COVID-19. [27]

This is now mentioned on page 2, lines 78-86 of the revised manuscript.

3)  Line 70: I would rephrase “vascular cells”, as at least for me it is not entirely clear what is meant here.

Authors’ response: Thank you for the comment. We have now revised the sentence (page 2, lines 91-93) :

Among others, vascular endothelial cells, platelets and blood cells are capable of secreting extracellular vesicles.[28,29]

4)  vWF is first mentioned in line 93, but the abbreviation comes in line 122 – this should be changed.

Authors’ response: Thank you for carefully reading our manuscript. We have now explained the abbreviation vWF after the first mention (page 2, line 68 of the revised manuscript).

Point-by-point response to Reviewer 2:  Thank you very much for carefully reading our manuscript and your valuable suggestions, which we have followed.

  1. p. 1,line 41: In discussing endothelial injury and microthrombosis with inflammation and complement activation as the primary pathology in acute COVID, in addition to references 9 and 13 should cite the initial report of such phenomena, as well as further follow-up  (i.e., Magro C, et al. Transl Res 2020;220:1-13 and Laurence J, et al. Am J Pathol 2022;192:1282-1294).[30]

Authors’ response: We have now included the mentioned references (reference 17 and 18 of the revised manuscript).

  1. P. 2, lines 75-76: In discussing the importance of tissue factor in COVID pathology should note induction of TF on microvessels by SARS-CoV-2 products (e.g., Subrahmanian S, et al. Thromb Haemostas 2021; 19:2268-2274.  

Authors’ response: Thank you for the comment. We now mention this on page 2, lines 97-103, page 3, 164-171 and on page 4, lines 184-186 with reference number 45 of the revised manuscript.

The other main component in thrombosis is the activation of the coagulation cascade either by tissue factor (TF; extrinsic pathway) or the intrinsic pathway.[4] Both pathways are active in COVID-19[16,19,15] and result in factor X activation causing thrombin generation.[4]

The up-regulation of tissue factor in different cells during COVID-19 including endothelial and epithelial cells, neutrophils, monocytes and macrophages as well as platelets links inflammation and thrombosis. [13-16] Tissue factor bearing extracellular vesicles are increasingly released into the circulation and activate the extrinsic coagulation pathway. [15] Extracellular vesicle tissue factor activity has been shown to be associated with thrombotic events and disease severity, circulating leukocytes and inflammatory markers, prothrombin time and d-dimer levels, vWF, ADAMTS13 (a disintegrin-like metallopeptidase with thrombospondin motif type 1 member 13), fibrinogen and plasmin–antiplasmin complexes.[15,17,18] In addition, contact activation of the coagulation system is triggered by neutrophil and complement activation. [19,15]

  1. P. 3, lines 113-114: It is stated that "early anticoagulation seems to be a sufficient treatment option," supported by references 49-53. However it should be noted that such a strategy based on heparin products is effective in non-critically ill hospitalized COVID patients (e.g., ref. 51) but that it is not effective in the critically ill hospitalized COVID patient (i.e., N Engl J Med 2021;385:777-789).

Authors’ response: Thank you for the valuable comment. We have now revised these statements in the manuscript:

Therapeutic anticoagulation using heparin seems to be most effective in non-critically ill COVID-19 patients,[31] but not in those with critical illness,[32] and further strategies to overcome the processes of immunothrombosis are needed. However, patient outcome with regard to different anticoagulation treatment regimens warrants further clinical trials. [31-33] In addition, impaired fibrinolysis has emerged as another challenging therapeutical aspect, requiring future research. [34-37] Interference with thrombin-induced pathways could also be of potential benefit, however, the occurrence of increased bleeding events might limit therapeutic success. [38,39] Moreover, anticoagulation regimens after recovery from COVID-19 to overcome the higher risk of deep vein thrombosis and pulmonary embolism are yet not well studied and should also consider a higher bleeding risk in the first months after SARS-CoV-2 infection. [40] (page 3, lines 146-157 of the revised manuscript)

  1. P. 4, line 155: As a corollary to point no. 3, should note that this "clinical benefit" with heparin is limited to certain categories of COVID patient.

Authors’ response: We have amended the sentence accordingly (page 4, lines 219-223 of the revised manuscript)

The clinical benefits of heparin administration in for e.g. non-critically ill COVID-19 patients [31] might also be explained by the prevention of histone-platelet interactions, which is achieved by high heparin concentrations and was shown to protect mice from histone-induced tissue damage, thrombocytopenia and death.[41]

  1. P. 5: At the end could mention that many of the endothelial-based pathologies in acute COVID-19 may extent to long COVID (e.g., Ahamed J, et al. J Clin Invest 2022;132:e161167).

Authors’ response: Thank you for the valuable comment. We have now included this into the revised manuscript:

The persistence of neutrophil activation and NET formation seems to be involved in the development of long COVID.[42] These changes can be observed even months after SARS-CoV-2 infection,[42] and viral persistence is suggested to trigger repeated inflammatory responses.[43] In post COVID associated changes neutrophils are discussed to promote fibrosis by protease release and to drive pulmonary symptoms.[42] In addition, NETosis is discussed to contribute to tumor growth, neurological disorders and worsening of concomitant diseases (like for e.g. hypertension or diabetes) after SARS-CoV-2 infection.[44]Given the involvement of NETs in (micro-)thrombosis and vasculitis,[45,46] it can be speculated that NETosis promotes post COVID endotheliopathy and pro-coagulability. Recently, there is increasing evidence for microthrombus formation as contributing factor to cardiopulmonary long COVID syndromes.[47] Moreover, as during acute SARS-CoV-2 infection, key factors of post COVID sequelae are based on endothelial injury, inflammation and pro-coagulability.[48,47] Virus-eliminating strategies might be helpful in the prevention of persistent COVID-associated complications. [49]

This is now mentioned on page 5, lines 276-290 of the revised manuscript.

In conclusion, endothelial dysfunction with glycocalyx disintegration can be regarded as initial trigger of SARS-CoV-2 mediated microthrombosis. Signaling pathways contributing to immunothrombosis are complex, involving innate and adaptive mechanisms. COVID-19-associated pro-thrombotic potential relies on well- published findings, such as endothelial cell infection[50] with glycocalyx destruction, promoting subsequent collagen exposure with platelet and coagulation activation.[51,52] Microthromboses upon platelet-platelet and platelet-leukocyte interactions further augment this response. [53-55] These are specific therapeutic targets to interfere with prolonged pro-coagulant and pro-thrombotic responses to viral infection. Evidence for long COVID treatment is still scarce, although many aspects of the endothelial-based pathologies of the acute SARS-CoV-2 infection may apply to the long-term sequelae. [47-49] Research on the elimination of viral persistence as well as interference with processes of immunothrombosis is still ongoing and merits further study.

This is now included in the conclusion on page 6, lines 314-327 of the revised manuscript.

  1. Virtually all of the material reviewed here has appeared in other reviews on acute COVID-19 pathophysiology. It would help for the authors to speculate at the end about potential types of intervention. Based on recent studies, could suggest use of anticoagulants with fibrinolytic activity such as argatroban, or use of specific anti-inflammatory agents or antivirals (e.g., based on the model for long COVID suggested in point no. 5).

Authors’ response:

Thank you for the comment.

COVID- specific therapeutic measures in the treatment of procoagulability and inflammation are not yet established. Prolonged anticoagulant therapy after COVID- infection might confer clinical benefits.[5,56] Impaired fibrinolysis was described in case reports and small clinical trials.[35,34,36,37] So far, an interventional trial is pending on the use of fibrinolytic drugs in these patients.

We discuss this on page 3, lines 146-157 and on page 4, lines 224-237/ page 5, lines 250-251 and 285-190:

Therapeutic anticoagulation using heparin seems to be most effective in non-critically ill COVID-19 patients,[31] but not in those with critical illness,[32] and further strategies to overcome the processes of immunothrombosis are needed. However, patient outcome with regard to different anticoagulation treatment regimens warrants further clinical trials. [31-33] In addition, impaired fibrinolysis has emerged as another challenging therapeutical aspect, requiring future research. [34-37] Interference with thrombin-induced pathways could also be of potential benefit, however, the occurrence of increased bleeding events might limit therapeutic success. [38,39] Moreover, anticoagulation regimens after recovery from COVID-19 to overcome the higher risk of deep vein thrombosis and pulmonary embolism are yet not well studied and should also consider a higher bleeding risk in the first months after SARS-CoV-2 infection. [40]

One of the linking elements between vWF and NET formation is possibly IL-6, having a central role in the cytokine release syndrome of COVID-19.[57,58] IL-6 is suggested to orchestrate a feedback mechanism between NETs and the vWF/ADAMTS13 axis.[57] The application of the IL-6 receptor antagonists tocilizumab and sarilumab showed to improve survival of critically-ill COVID-19 patients.[59] In addition, NETosis is triggered by other cytokines, such as for e.g. IL- 1ß and IL-8. [60,44] Blockage of IL-1ß by canakinumab was recently tested in patients with SARS-CoV-2 infection.[61] The ‘Canakinumab in Patients With COVID-19 and Type 2 Diabetes’ (CanCovDia) trial included 116 diabetic patients with COVID-19, which were randomized into two groups with canakinumab vs. placebo in addition to standard treatment.[61] Although canakinumab did not result in an improvement of survival, length of ICU stay or hospitalization as well as ventilation time and glycemic control was more rapidly achieved requiring less treatment with anti-diabetic drugs.[61] As shown in this trial, blocking of IL-1ß might be of most advantage for patients at high risk for severe COVID-19 disease course, including those with diabetes-related complications, kidney failure and adiposity.[61]

Recently, there is increasing evidence for microthrombus formation as contributing factor to cardiopulmonary long COVID syndromes.[47] Moreover, as during acute SARS-CoV-2 infection, key factors of post COVID sequelae are based on endothelial injury, inflammation and pro-coagulability.[48,47] Virus-eliminating strategies might be helpful in the prevention of persistent COVID-associated complications. [49]

  1. Fig. 1: In my review copy, the labels in every circle or box are highlighted by "Q" marks. Is this an artefact of a Word processing program? Do these marks represent citations?

Authors’ response:

We apologize for this difficulty and thank you for the remark. Indeed, it has to be an artefact, which should be considered during further article processing.

References:

  1. Constantinescu-Bercu, A., Wang, Y. A., Woollard, K. J., Mangin, P., Vanhoorelbeke, K., Crawley, J. T. B., et al. (2021). The GPIbα intracellular tail - role in transducing VWF- and collagen/GPVI-mediated signaling. Haematologica, 107(4), 933-946, doi:10.3324/haematol.2020.278242.
  2. Sadler, J. E. (1991). von Willebrand factor. Journal of Biological Chemistry, 266(34), 22777-22780, doi:https://doi.org/10.1016/S0021-9258(18)54418-5.
  3. Philippe, A., Gendron, N., Bory, O., Beauvais, A., Mirault, T., Planquette, B., et al. (2021). Von Willebrand factor collagen-binding capacity predicts in-hospital mortality in COVID-19 patients: insight from VWF/ADAMTS13 ratio imbalance. Angiogenesis, 24(3), 407-411, doi:10.1007/s10456-021-09789-3.
  4. Andrade, S. A. d., de Souza, D. A., Torres, A. L., de Lima, C. F. G., Ebram, M. C., Celano, R. M. G., et al. (2022). Pathophysiology of COVID-19: Critical Role of Hemostasis. [Review]. Frontiers in Cellular and Infection Microbiology, 12.
  5. Li, R., & Emsley, J. (2013). The organizing principle of the platelet glycoprotein Ib–IX–V complex. Journal of Thrombosis and Haemostasis, 11(4), 605-614, doi:https://doi.org/10.1111/jth.12144.
  6. Into, T., Kanno, Y., Dohkan, J.-i., Nakashima, M., Inomata, M., Shibata, K.-i., et al. (2007). Pathogen Recognition by Toll-like Receptor 2 Activates Weibel-Palade Body Exocytosis in Human Aortic Endothelial Cells*. Journal of Biological Chemistry, 282(11), 8134-8141, doi:https://doi.org/10.1074/jbc.M609962200.
  7. Singh, B., Biswas, I., Bhagat, S., Surya Kumari, S., & Khan, G. A. (2016). HMGB1 facilitates hypoxia-induced vWF upregulation through TLR2-MYD88-SP1 pathway. [https://doi.org/10.1002/eji.201646386]. European Journal of Immunology, 46(10), 2388-2400, doi:https://doi.org/10.1002/eji.201646386.
  8. Carestia, A., Kaufman, T., Rivadeneyra, L., Landoni, V. I., Pozner, R. G., Negrotto, S., et al. (2016). Mediators and molecular pathways involved in the regulation of neutrophil extracellular trap formation mediated by activated platelets. [https://doi.org/10.1189/jlb.3A0415-161R]. Journal of Leukocyte Biology, 99(1), 153-162, doi:https://doi.org/10.1189/jlb.3A0415-161R.
  9. Choudhury, A., & Mukherjee, S. (2020). In silico studies on the comparative characterization of the interactions of SARS-CoV-2 spike glycoprotein with ACE-2 receptor homologs and human TLRs. [https://doi.org/10.1002/jmv.25987]. Journal of Medical Virology, 92(10), 2105-2113, doi:https://doi.org/10.1002/jmv.25987.
  10. Fard, M. B., Fard, S. B., Ramazi, S., Atashi, A., & Eslamifar, Z. (2021). Thrombosis in COVID-19 infection: Role of platelet activation-mediated immunity. Thrombosis Journal, 19(1), 59, doi:10.1186/s12959-021-00311-9.
  11. Teixeira, P. C., Dorneles, G. P., Santana Filho, P. C., da Silva, I. M., Schipper, L. L., Postiga, I. A. L., et al. (2021). Increased LPS levels coexist with systemic inflammation and result in monocyte activation in severe COVID-19 patients. International Immunopharmacology, 100, 108125, doi:https://doi.org/10.1016/j.intimp.2021.108125.
  12. Clark, S. R., Ma, A. C., Tavener, S. A., McDonald, B., Goodarzi, Z., Kelly, M. M., et al. (2007). Platelet TLR4 activates neutrophil extracellular traps to ensnare bacteria in septic blood. Nature Medicine, 13(4), 463-469, doi:10.1038/nm1565.
  13. Nguyen, D., Jeon, H.-M., & Lee, J. (2022). Tissue factor links inflammation, thrombosis, and senescence in COVID-19. Scientific Reports, 12(1), 19842, doi:10.1038/s41598-022-23950-y.
  14. Sachetto, T. A. A., & Mackman, N. (2022). Tissue Factor and COVID-19: An Update. Current Drug Targets, 23(17), 1573-1577, doi:http://dx.doi.org/10.2174/1389450123666220926144432.
  15. Gorog, D. A., Storey, R. F., Gurbel, P. A., Tantry, U. S., Berger, J. S., Chan, M. Y., et al. (2022). Current and novel biomarkers of thrombotic risk in COVID-19: a Consensus Statement from the International COVID-19 Thrombosis Biomarkers Colloquium. Nature Reviews Cardiology, 19(7), 475-495, doi:10.1038/s41569-021-00665-7.
  16. Subrahmanian, S., Borczuk, A., Salvatore, S., Fung, K.-M., Merrill, J. T., Laurence, J., et al. (2021). Tissue factor upregulation is associated with SARS-CoV-2 in the lungs of COVID-19 patients. [https://doi.org/10.1111/jth.15451]. Journal of Thrombosis and Haemostasis, 19(9), 2268-2274, doi:https://doi.org/10.1111/jth.15451.
  17. Rosell, A., Havervall, S., von Meijenfeldt, F., Hisada, Y., Aguilera, K., Grover, S. P., et al. (2021). Patients With COVID-19 Have Elevated Levels of Circulating Extracellular Vesicle Tissue Factor Activity That Is Associated With Severity and Mortality—Brief Report. Arteriosclerosis, Thrombosis, and Vascular Biology, 41(2), 878-882, doi:10.1161/ATVBAHA.120.315547.
  18. Guervilly, C., Bonifay, A., Burtey, S., Sabatier, F., Cauchois, R., Abdili, E., et al. (2021). Dissemination of extreme levels of extracellular vesicles: tissue factor activity in patients with severe COVID-19. Blood Advances, 5(3), 628-634, doi:10.1182/bloodadvances.2020003308.
  19. Busch, M. H., Timmermans, S. A. M. E. G., Nagy, M., Visser, M., Huckriede, J., Aendekerk, J. P., et al. (2020). Neutrophils and Contact Activation of Coagulation as Potential Drivers of COVID-19. Circulation, 142(18), 1787-1790, doi:10.1161/CIRCULATIONAHA.120.050656.
  20. Gremmel, T., Frelinger, A. L., 3rd, & Michelson, A. D. (2016). Platelet Physiology. Semin Thromb Hemost, 42(3), 191-204, doi:10.1055/s-0035-1564835.
  21. Parker, W. A. E., & Storey, R. F. (2016). Long-term antiplatelet therapy following myocardial infarction: implications of PEGASUS-TIMI 54. Heart, 102(10), 783, doi:10.1136/heartjnl-2015-307858.
  22. Storey, R. F., Sanderson, H. M., White, A. E., May, J. A., Cameron, K. E., & Heptinstall, S. (2000). The central role of the P2T receptor in amplification of human platelet activation, aggregation, secretion and procoagulant activity. [https://doi.org/10.1046/j.1365-2141.2000.02208.x]. British Journal of Haematology, 110(4), 925-934, doi:https://doi.org/10.1046/j.1365-2141.2000.02208.x.
  23. Storey, R. F., Judge, H. M., Wilcox, R. G., & Heptinstall, S. (2002). Inhibition of ADP-induced P-selectin Expression and Platelet-Leukocyte Conjugate Formation by Clopidogrel and the P2Y12 Receptor Antagonist AR-C69931MX but not Aspirin. Thromb Haemost, 88(09), 488-494.
  24. Etulain, J., Martinod, K., Wong, S. L., Cifuni, S. M., Schattner, M., & Wagner, D. D. (2015). P-selectin promotes neutrophil extracellular trap formation in mice. Blood, 126(2), 242-246, doi:10.1182/blood-2015-01-624023.
  25. Wagner, D. D., & Heger, L. A. (2022). Thromboinflammation: From Atherosclerosis to COVID-19. Arteriosclerosis, Thrombosis, and Vascular Biology, 42(9), 1103-1112, doi:10.1161/ATVBAHA.122.317162.
  26. Michelson, A. D., Barnard, M. R., Hechtman, H. B., MacGregor, H., Connolly, R. J., Loscalzo, J., et al. (1996). In vivo tracking of platelets: circulating degranulated platelets rapidly lose surface P-selectin but continue to circulate and function. Proceedings of the National Academy of Sciences, 93(21), 11877-11882, doi:10.1073/pnas.93.21.11877.
  27. Garcia, C., Au Duong, J., Poëtte, M., Ribes, A., Payre, B., Mémier, V., et al. (2022). Platelet activation and partial desensitization are associated with viral xenophagy in patients with severe COVID-19. Blood Advances, 6(13), 3884-3898, doi:10.1182/bloodadvances.2022007143.
  28. Brahmer, A., Neuberger, E., Esch-Heisser, L., Haller, N., Jorgensen, M. M., Baek, R., et al. (2019). Platelets, endothelial cells and leukocytes contribute to the exercise-triggered release of extracellular vesicles into the circulation. Journal of Extracellular Vesicles, 8(1), 1615820, doi:10.1080/20013078.2019.1615820.
  29. Nederveen, J. P., Warnier, G., Di Carlo, A., Nilsson, M. I., & Tarnopolsky, M. A. (2021). Extracellular Vesicles and Exosomes: Insights From Exercise Science. [Review]. Frontiers in Physiology, 11.
  30. Laurence, J., Nuovo, G., Racine-Brzostek, S. E., Seshadri, M., Elhadad, S., Crowson, A. N., et al. (2022). Premortem Skin Biopsy Assessing Microthrombi, Interferon Type I Antiviral and Regulatory Proteins, and Complement Deposition Correlates with Coronavirus Disease 2019 Clinical Stage. The American Journal of Pathology, 192(9), 1282-1294, doi:https://doi.org/10.1016/j.ajpath.2022.05.006.
  31. Therapeutic Anticoagulation with Heparin in Noncritically Ill Patients with Covid-19 (2021). N Engl J Med, 385(9), 790-802, doi:10.1056/NEJMoa2105911.
  32. Therapeutic Anticoagulation with Heparin in Critically Ill Patients with Covid-19 (2021). New England Journal of Medicine, 385(9), 777-789, doi:10.1056/NEJMoa2103417.
  33. Ionescu, F., Jaiyesimi, I., Petrescu, I., Lawler, P. R., Castillo, E., Munoz-Maldonado, Y., et al. (2021). Association of anticoagulation dose and survival in hospitalized COVID-19 patients: A retrospective propensity score-weighted analysis. [https://doi.org/10.1111/ejh.13533]. European Journal of Haematology, 106(2), 165-174, doi:https://doi.org/10.1111/ejh.13533.
  34. Bakchoul, T., Hammer, S., Lang, P., & Rosenberger, P. (2020). Fibrinolysis shut down in COVID-19 patients: Report on two severe cases with potential diagnostic and clinical relevance. Thrombosis Update, 1, 100008, doi:https://doi.org/10.1016/j.tru.2020.100008.
  35. Wygrecka, M., Birnhuber, A., Seeliger, B., Michalick, L., Pak, O., Schultz, A.-S., et al. (2022). Altered fibrin clot structure and dysregulated fibrinolysis contribute to thrombosis risk in severe COVID-19. Blood Advances, 6(3), 1074-1087, doi:10.1182/bloodadvances.2021004816.
  36. Ji, H.-L., Dai, Y., & Zhao, R. (2022). Fibrinolytic therapy for COVID-19: a review of case series. Acta Pharmacologica Sinica, 43(8), 2168-2170, doi:10.1038/s41401-021-00827-w.
  37. Zuo, Y., Warnock, M., Harbaugh, A., Yalavarthi, S., Gockman, K., Zuo, M., et al. (2021). Plasma tissue plasminogen activator and plasminogen activator inhibitor-1 in hospitalized COVID-19 patients. Scientific Reports, 11(1), 1580, doi:10.1038/s41598-020-80010-z.
  38. Rovai, E. S., Alves, T., & Holzhausen, M. (2020). Protease-activated receptor 1 as a potential therapeutic target for COVID-19. Experimental Biology and Medicine, 246(6), 688-694, doi:10.1177/1535370220978372.
  39. Pultar, J., Wadowski, P. P., Panzer, S., & Gremmel, T. (2019). Oral antiplatelet agents in cardiovascular disease. VASA, 48(4), 291-302, doi:10.1024/0301-1526/a000753.
  40. Katsoularis, I., Fonseca-Rodríguez, O., Farrington, P., Jerndal, H., Lundevaller, E. H., Sund, M., et al. (2022). Risks of deep vein thrombosis, pulmonary embolism, and bleeding after covid-19: nationwide self-controlled cases series and matched cohort study. BMJ, 377, e069590, doi:10.1136/bmj-2021-069590.
  41. Fuchs, T. A., Bhandari, A. A., & Wagner, D. D. (2011). Histones induce rapid and profound thrombocytopenia in mice. Blood, 118(13), 3708-3714, doi:10.1182/blood-2011-01-332676.
  42. George, P. M., Reed, A., Desai, S. R., Devaraj, A., Faiez, T. S., Laverty, S., et al. A persistent neutrophil-associated immune signature characterizes post–COVID-19 pulmonary sequelae. Science Translational Medicine, 14(671), eabo5795, doi:10.1126/scitranslmed.abo5795.
  43. Mantovani, A., Morrone, M. C., Patrono, C., Santoro, M. G., Schiaffino, S., Remuzzi, G., et al. (2022). Long Covid: where we stand and challenges ahead. Cell Death & Differentiation, 29(10), 1891-1900, doi:10.1038/s41418-022-01052-6.
  44. Zhu, Y., Chen, X., & Liu, X. (2022). NETosis and Neutrophil Extracellular Traps in COVID-19: Immunothrombosis and Beyond. [Mini Review]. Frontiers in Immunology, 13.
  45. Thierry, A. R., & Roch, B. (2020). Neutrophil Extracellular Traps and By-Products Play a Key Role in COVID-19: Pathogenesis, Risk Factors, and Therapy. Journal of Clinical Medicine, 9(9). doi:10.3390/jcm9092942
  46. Middleton, E. A., He, X.-Y., Denorme, F., Campbell, R. A., Ng, D., Salvatore, S. P., et al. (2020). Neutrophil extracellular traps contribute to immunothrombosis in COVID-19 acute respiratory distress syndrome. Blood, 136(10), 1169-1179, doi:10.1182/blood.2020007008.
  47. Ahamed, J., & Laurence, J. (2022). Long COVID endotheliopathy: hypothesized mechanisms and potential therapeutic approaches. The Journal of Clinical Investigation, 132(15), doi:10.1172/JCI161167.
  48. Wang, C., Yu, C., Jing, H., Wu, X., Novakovic, V. A., Xie, R., et al. (2022). Long COVID: The Nature of Thrombotic Sequelae Determines the Necessity of Early Anticoagulation. [Review]. Frontiers in Cellular and Infection Microbiology, 12.
  49. Xie, Y., Choi, T., & Al-Aly, Z. (2022). Nirmatrelvir and the Risk of Post-Acute Sequelae of COVID-19. medRxiv, 2022.2011.2003.22281783, doi:10.1101/2022.11.03.22281783.
  50. Varga, Z., Flammer, A. J., Steiger, P., Haberecker, M., Andermatt, R., Zinkernagel, A. S., et al. (2020). Endothelial cell infection and endotheliitis in COVID-19. The Lancet, 395(10234), 1417-1418, doi:10.1016/S0140-6736(20)30937-5.
  51. Root-Bernstein, R., Huber, J., & Ziehl, A. (2022). Complementary Sets of Autoantibodies Induced by SARS-CoV-2, Adenovirus and Bacterial Antigens Cross-React with Human Blood Protein Antigens in COVID-19 Coagulopathies. International Journal of Molecular Sciences, 23(19). doi:10.3390/ijms231911500
  52. Xu, S.-w., Ilyas, I., & Weng, J.-p. (2022). Endothelial dysfunction in COVID-19: an overview of evidence, biomarkers, mechanisms and potential therapies. Acta Pharmacologica Sinica, doi:10.1038/s41401-022-00998-0.
  53. Parker, W. A. E., Orme, R. C., Hanson, J., Stokes, H. M., Bridge, C. M., Shaw, P. A., et al. (2019). Very-low-dose twice-daily aspirin maintains platelet inhibition and improves haemostasis during dual-antiplatelet therapy for acute coronary syndrome. Platelets, 30(2), 148-157, doi:10.1080/09537104.2019.1572880.
  54. Paul, B. Z., Jin, J., & Kunapuli, S. P. (1999). Molecular mechanism of thromboxane A(2)-induced platelet aggregation. Essential role for p2t(ac) and alpha(2a) receptors. J Biol Chem, 274(41), 29108-29114, doi:10.1074/jbc.274.41.29108.
  55. Caillon, A., Trimaille, A., Favre, J., Jesel, L., Morel, O., & Kauffenstein, G. (2022). Role of neutrophils, platelets, and extracellular vesicles and their interactions in COVID-19-associated thrombopathy. [https://doi.org/10.1111/jth.15566]. Journal of Thrombosis and Haemostasis, 20(1), 17-31, doi:https://doi.org/10.1111/jth.15566.
  56. Chandra, A., Chakraborty, U., Ghosh, S., & Dasgupta, S. (2022). Anticoagulation in COVID-19: current concepts and controversies. Postgraduate Medical Journal, 98(1159), 395, doi:10.1136/postgradmedj-2021-139923.
  57. Fernández-Pérez, M. P., Águila, S., Reguilón-Gallego, L., de los Reyes-García, A. M., Miñano, A., Bravo-Pérez, C., et al. (2021). Neutrophil extracellular traps and von Willebrand factor are allies that negatively influence COVID-19 outcomes. [https://doi.org/10.1002/ctm2.268]. Clinical and Translational Medicine, 11(1), e268, doi:https://doi.org/10.1002/ctm2.268.
  58. Moore, J. B., & June, C. H. (2020). Cytokine release syndrome in severe COVID-19. Science, 368(6490), 473-474, doi:10.1126/science.abb8925.
  59. Interleukin-6 Receptor Antagonists in Critically Ill Patients with Covid-19 (2021). New England Journal of Medicine, 384(16), 1491-1502, doi:10.1056/NEJMoa2100433.
  60. Thiam, H. R., Wong, S. L., Wagner, D. D., & Waterman, C. M. (2020). Cellular Mechanisms of NETosis. Annual Review of Cell and Developmental Biology, 36(1), 191-218, doi:10.1146/annurev-cellbio-020520-111016.
  61. Hepprich, M., Mudry, J. M., Gregoriano, C., Jornayvaz, F. R., Carballo, S., Wojtusciszyn, A., et al. (2022). Canakinumab in patients with COVID-19 and type 2 diabetes – A multicentre, randomised, double-blind, placebo-controlled trial. eClinicalMedicine, 53, doi:10.1016/j.eclinm.2022.101649.

Reviewer 2 Report

1. p. 1,line 41: In discussing endothelial injury and microthrombosis with inflammation and complement activation as the primary pathology in acute COVID, in addition to references 9 and 13 should cite the initial report of such phenomena, as well as further follow-up  (i.e., Magro C, et al. Transl Res 2020;220:1-13 and Laurence J, et al. Am J Pathol 2022;192:1282-1294).

2. P. 2, lines 75-76: In discussing the importance of tissue factor in COVID pathology should note induction of TF on microvessels by SARS-CoV-2 products (e.g., Subrahmanian S, et al. Thromb Haemostas 2021; 19:2268-2274.  

3. P. 3, lines 113-114: It is stated that "early anticoagulation seems to be a sufficient treatment option," supported by references 49-53. However it should be noted that such a strategy based on heparin products is effective in non-critically ill hospitalized COVID patients (e.g., ref. 51) but that it is not effective in the critically ill hospitalized COVID patient (i.e., N Engl J Med 2021;385:777-789).

4. P. 4, line 155: As a corollary to point no. 3, should note that this "clinical benefit" with heparin is limited to certain categories of COVID patient.

5. P. 5: At the end could mention that many of the endothelial-based pathologies in acute COVID-19 may extent to long COVID (e.g., Ahamed J, et al. J Clin Invest 2022;132:e161167).

6. Virtually all of the material reviewed here has appeared in other reviews on acute COVID-19 pathophysiology. It would help for the authors to speculate at the end about potential types of intervention. Based on recent studies, could suggest use of anticoagulants with fibrinolytic activity such as argatroban, or use of specific anti-inflammatory agents or antivirals (e.g., based on the model for long COVID suggested in point no. 5).

7. Fig. 1: In my review copy, the labels in every circle or box are highlighted by "Q" marks. Is this an artefact of a Word processing program? Do these marks represent citations?

Author Response

(The authors gave the same response as above.)

Round 2

Reviewer 1 Report

My comments have been addressed.

Reviewer 2 Report

The authors have responded appropriately to all of my concerns.